In vitro biomechanical evaluation of internal fixation techniques on the canine lumbosacral junction

Early Peter 1 pjearly@ncsu.edu
Mente Peter 2
Dillard Stacy 3
Roe Simon 1
1 College of Veterinary Medicine, North Carolina State University , Raleigh, NC , USA
2 Joint Department of Biomedical Engineering at the University of North Carolina Chapel Hill, North Carolina State University , Raleigh, NC , USA
3 Veterinary Neurology Center , Tustin, CA , USA
Esteban María Ángeles
Electronic publication date: 2015 Aug 20
Publication date: 2015
Volume: 3
Electronic Location ID: e1094
Received 2015 Mar 17; Accepted 2015 Jun 18
Copyright: © 2015 Early et al.
Copyright year: 2015
Copyright holder: Early et al.
License: This is an open access article distributed under the terms of the Creative Commons Attribution License, which permits unrestricted use, distribution, reproduction and adaptation in any medium and for any purpose provided that it is properly attributed. For attribution, the original author(s), title, publication source (PeerJ) and either DOI or URL of the article must be cited.
License URL: https://creativecommons.org/licenses/by/4.0/

Keywords: Lumbosacral, Biomechanics, Dog, SOP, Transarticular facet, PMMA

Funding: The Department of Clinical Sciences at NCSU supported part of the funding for this system. The funders had no role in study design, data collection and analysis, decision to publish, or preparation of the manuscript.

==============================
Few biomechanical studies have evaluated the effect of internal stabilization techniques after decompressive surgery on the stability of the canine lumbosacral junction. The purpose of this canine cadaver study is to evaluate the stability of the canine lumbosacral (LS) spine in flexion and extension following laminectomy and discectomy and then stabilization with each of the three techniques: pins and polymethylmethacrylate (P/PMMA), two dorsal locking plates (SOP) or bilateral transarticular facet screws (FACET).Using a cantilever biomechanical system, bending moments were applied to the LS and range of motion (ROM) was recorded via a rotational potentiometer. With 3 Nm, the ROM (n = 4 in each group) for P/PMMA, SOP and FACET were 1.92 ± 0.96°, 2.56 ± 0.55°and 3.18 ± 1.14°, respectively. With moments up to 35 Nm, the P/PMMA specimens appeared stable. Sacroiliac motion in the SOP and FACET groups invalidated further comparisons. Each of the stabilization techniques (P/PMMA, SOP, and FACET) significantly decreased the range of motion in flexion and extension for low bending moments.

Introduction

Degenerative Lumbosacral Stenosis (DLS) is a common cause of caudal lumbar pain, difficulty in sitting and difficulty rising in middle aged large breed dogs (Meij & Bergknut, 2010). DLS is commonly associated with Hansen type II disc degeneration, ligamentous hypertrophy, articular facet and joint capsule hypertrophy, spondylosis deformans, subluxation of the sacrum and lumbosacral instability. It is thought that increased motion at the lumbosacral junction is the most important contributor to the degenerative changes and progression of clinical signs in dogs (Meij & Bergknut, 2010).

Surgical management is recommended for patients with severe or recurrent pain that is not responsive to medical management or when neurologic deficits are present (Johnston & Tobias, 2012a). Common surgical options for DLS include dorsal laminectomy alone or in combination with a partial discectomy, dorsal laminectomy combined with fixation and fusion or lateral foraminotomy (Meij et al., 2007; Hankin et al., 2012; Smolders et al., 2012a; Smolders et al., 2012b). Two previous biomechanical studies have shown that a dorsal laminectomy with partial discectomy increases lumbosacral movement, which may lead to instability (Smolders et al., 2012a; Early et al., 2013). Some surgeons feel that dorsal stabilization is indicated to provide stability to the LS junction. If instability is present, then dorsal stabilization will limit the range of motion. Even in the absence of significant instability, if there is nerve impingement secondary to proliferation of tissue around the LS junction, rigid fixation may reduce the intensity of the fibrous response, thus relieving the pressure on the spinal nerves. An optimal configuration or system of fixation has yet to be determined. Two of the most widely used and clinically accepted fixation techniques are (1) positive profile threaded pins and polymethylmethacrylate (Weh & Kraus, 2007) and (2) bilateral transarticular facet screw stabilization (Hankin et al., 2012). The SOP™ locking plate system may also be suitable, enabling screws to be directed into the limited bone available (Johnston & Tobias, 2012b).

Currently there remains inadequate cased-based evidence to support the use of surgical intervention over conservative management for DLS (Jeffery, Barker & Harcourt-Brown, 2014). There seems to be major controversy with regard to the type of surgery that may be chosen as well. Most veterinarians would support surgical intervention in dogs with severe pain and fecal or urinary incontinence. Of the common surgical options listed above, none is without limitations. Dorsal laminectomy and discectomy may not alleviate compression of the L7 nerve in the foramen. Foraminotomy alone does not allow for removal of protruding disc and ligamentous compression within the spinal canal. Combining a dorsal laminectomy and foraminotomy may increase the risk of articular facet fracture. Stabilization of the LS junction is performed when the goal is to reduce the dynamic components of nerve compression within the vertebral canal, or when it is the surgeon’s opinion that the LS junction is unstable. Potential problems with the LS stabilization techniques include poor implant placement (i.e., implants within the vertebral canal) and inability to assure long term rigid fixation and bony fusion (Smolders et al., 2012b).

The purpose of this canine cadaver study was to evaluate the range of flexion and extension and load to failure of the canine lumbosacral spine following stabilization with pins and polymethylmethacrylate (P/PMMA), two dorsal locking plates (SOP) and bilateral transarticular facet screws (FACET), after a dorsal laminectomy and partial discectomy.

Materials and Methods

The caudal lumbar spine (L6-7), sacrum and pelvis were harvested from 12 euthanized skeletally mature non-chondrodystrophic dogs with weights ranging from 23.6 to 36.7 kg (median 30.4 kg). Specimens were radiographed and no degenerative changes were noted. The pelvis was fixed in a resin mold (Bondo, Bondo Corporation, Atlanta, Georgia, USA), which was mounted to the base of a testing machine (MTS, Canton, Massachusetts, USA). An eyebolt screwed into the center of L6 was attached to the actuator so that the spine segment could be flexed and extended. The applied moment (Nm) was calculated by multiplying the applied load (N) by the distance from the LS space to the actuator (m). Movement of L7 was monitored by a weighted rotational potentiometer (P1411, Novotechnik, Southborough, Massachusetts, USA) attached to the ventral aspect of the vertebra. In a previously reported study, the specimens were conditioned at ±1.5 Nm, at a rate of two cycles per second and range of motion (ROM) were measured for ±3 Nm of bending (Early et al., 2013). An L7-S1 dorsal laminectomy and partial discectomy was performed and the ROM measured. The ROM of the intact specimens was 32.8 ± 6.4°and, after laminectomy and discectomy, this increased to 40.2 ± 5.6° (Early et al., 2013).

Following the ROM analysis, one of the three fixation techniques (P/PMMA, SOP and FACET) was applied to each of the specimens. There were 4 specimens per group. Implant entry points for the three fixation techniques are identified in Fig. 1 and radiographs depicting each technique are given in Fig. 2. The P/PMMA construct consisted of six positive profile 4.0 mm (5/32″) external fixation half-pins (Interface™, Imex Veterinary, Inc., Longveiw, Texas, USA). Two pins were placed into the pedicle of L7, two pins into the sacrum and two pins into the ilium. Predrilling for pins was not performed. PMMA was applied, incorporating all of the pins (Fossum, 1997). The PMMA was contoured and in close contact with the bone surface of L7 and sacrum. The PMMA was allowed to harden for a minimum of 1 h before testing.

Figure 1 Implants points of entry into L7, Sacrum and Ilium.

Dorsal view of the skeletal structures of the canine lumbosacral junction, showing the points of entry of the implants into L7, Sacrum and Ilium. The external fixation pins (for the P/PMMA) entry points are denoted with open dark gray circles with cross marks in the middle. The SOP locking plate entry points are denoted by solid black circles and the bilateral transarticular facet screws entry points and directions are denoted by black arrows.

Figure 2 Postoperative radiographs of the three stabilization techniques.

Postoperative radiographs, lateral and dorsoventral, of the three stabilization techniques. (A) External fixation pins and PMMA, (B) SOP™ Locking Plate System and (C) Bilateral transarticular facet screws.

The SOP fixation consisted of two 5-hole 3.5 mm locking plates (SOP™, Orthomed Ltd., West Yorkshire, UK) that were positioned parallel on either side dorsolaterally and secured to the pedicle of L7 and sacrum with two 3.5 mm cortical screws (Depuy Synthes Vet, West Chester, Pennsylvania, USA) in each plate. All of the screws for the SOP construct were placed in the most cranial pearl (hole 1) skipping the second pearl and then placing the second screw in the third pearl (hole 3).

The FACET fixation consisted of two 3.5 mm cortical screws oriented from the dorsal articular processes of L7, into the sacrum using a positional technique (Sharp & Wheeler, 2005).

After each fixation technique was applied, the specimen was preconditioned at ±1.5 Nm for 5 cycles, then loaded at ±3 Nm for 5 cycles to measure ROM. Subsequently, the stabilized specimens were subject to an incrementally increasing load, starting at ±2.5 Nm and increasing by 2.5 Nm after each set of 5 loading cycles, until testing was concluded. Testing was concluded if: (1) motion of L7 was greater than 10° in flexion or extension, (2) implant failure or bony fracture occurred; or (3) a bending moment of 35 Nm was applied (Smith et al., 2004). After all ROM testing was complete, lateral and dorsoventral radiographs were made of all specimens and the failure mechanism evaluated on these and on the specimens.

The ROM with ±3.5 Nm applied moment was compared between the stabilized specimens and the intact and decompressed data available from a previous study (Early et al., 2013). If differences were identified using ANOVA, individual comparisons were made using the least squares means test, and an overall P value of 0.05 to determine significance (SAS v9.1.3 Service pack 4, SAS Institute Inc., Cary, North Carolina, USA). Because of issues identified during the incremental load to failure study, statistical comparison of load to failure data was not performed.

Results

The ROM with ±3.5 Nm for the P/PMMA, SOP and FACET techniques were 1.92 ± 0.96°, 2.56 ± 0.55°, and 3.18 ± 1.14°, respectively, Fig. 3. After each fixation technique was applied the ROM of the stabilized specimens was significantly decreased (p < 0.001) compared to ROM after dorsal laminectomy and discectomy (mean of all specimens for all three groups =40.2 ± 5.6° (Early et al., 2013). One of the FACET specimens failed because of fracture around the screw with 14.1 Nm applied while in extension. One of the SOP specimens failed by loosening of the screws in L7 with 12.7 Nm applied while in extension. The other three specimens in each of the FACET and SOP groups failed because L7 motion was greater than 10°, though most of that motion originated at the SI joints. There was no failure of the fixation noted on gross inspection, or on radiographs. In the P/PMMA group, testing was stopped at 35 Nm of bending for three specimens, with no implant failure noted on gross inspection, or on radiographs. In the other P/PMMA specimen, the eyebolt fractured through L6 when a 25 Nm moment was applied.

Figure 3 Typical load–deflection curve in a canine cadaver lumbosacral spine during cyclic loading (flexion and extension) of spines after dorsal laminectomy and partial discectomy (DL) and each stabilization technique (SOP—black solid, FACET—light grey solid and P/PMMA—dark grey dashed).

Range of motion (ROM) was the L7 angulation change between flexion and extension with 3 Nm of bending moment applied.

Discussion

This study demonstrates that the LS region had much less range of motion after stabilization with each fixation technique, but, because the P/PMMA technique bridges the SI joint, and the FACET and SOP techniques did not, the specimens moved very differently during testing at higher bending moments. For this reason, we felt that it was not appropriate to make direct comparisons of failure using the mechanical data.

Visually and mechanically, the P/PMMA technique appeared to provide good stability. Three of four specimens resisted the highest applied moment with no evidence of failure. As the flexion and extension moment was increased on the FACET and SOP specimens, there was increasing motion of the SI joint. As this allowed L7 to move relative to the pelvis, this motion was included in the data. Movement of L7 relative to S1 could not be separated from the SI motion.

Visually, no motion was observed in the SOP specimens, except in one, where one screw loosened in L7. This suggests that, in this configuration and testing mode, the screw-bone interface was the weaker element. This may be due to slightly incorrect placement of this particular screw, or to the fact that there were only 2 screws in each vertebrae. There is a very narrow region into which the screw is inserted in L7 to optimize its purchase, while not damaging adjacent structures (Smolders et al., 2012b). The recently developed SOP™ Locking Plate System combines the advantages of a fixed angle stabilization system, like the P/PMMA, with lower bulk. The SOP™ plate can be contoured so that the locked screws are directed into the limited available bone stock (Johnston & Tobias, 2012b). When using the SOP™ locking plate system in the lumbar and sacral vertebrae, the following guidelines have been recommended recently—use SOP locking plates bilaterally, twist and contour the SOP caudally to engage the iliac shaft, recommend 4 screws but a minimum of 3 screws in each vertebral body, use the longest possible cortical screws to engage maximum amount of vertebral bone and have the SOP plate as close to the bone as possible while avoiding damage to emerging nerve roots (Orthomed). The configuration used in this study was selected before the above recommendations were available. The 2-screw configuration was chosen to mimic the pedicle rod and screw stabilization technique commonly used in humans and by some surgeons in dogs (Smolders et al., 2012b).

In the FACET specimens, slight motion was apparent between the facets. Fracture of the facet because it is weakened by the screw, as occurred in one specimen, is a known potential complication of this technique (Sharp & Wheeler, 2005; Hankin et al., 2012).

Previous in vitro cadaver studies have evaluated the biomechanical effects of stabilization after concurrent dorsal laminectomy and partial discectomy on the lumbosacral junction in the dog have yielded similar results (Benninger et al., 2004; Meij et al., 2007; Smolders et al., 2012c). In the Meij et al., 2007 study, a pedicle screw and rod fixation significantly stabilized the lumbosacral spine by decreasing the ROM from 29.1 ± 5.60° to 11.7 ± 3.30°. In the Smolders et al., 2012c study, a nucleus pulposus prosthesis effectively decreased the ROM of the lumbosacral spine by 8.8%. These studies evaluated the lumbosacral spine segments using 4-point bending. In this study, a cantilever system was used as it was easier to instrument and load the spinal segment. While the applied moment varies over the length of the specimen in cantilever bending, the moment applied to the LS articulation is easily calculated.

This study was intended to evaluate clinically accepted techniques for stabilization of the LS junction. Several distinctions should be noted as these potentially alter and give significant advantages to the various biomechanics of each construct. The P/PMMA fixation has the potential advantage of six screws and points of fixation, while the SOP and FACET have four and two points of fixation respectively. The P/PMMA constructs used 4.0 mm positive profile pins with a 3.2 mm shaft, while the SOP and FACET constructs used 3.5 mm cortical screws, with a 2.4 mm shaft. The P/PMMA construct was thicker, bulkier and in more intimate contact with the L7 and sacral vertebrae providing a buttress stabilization, which likely contributed to the more stable appearance of this group.

An interesting finding of this study was that the motion at the sacroiliac (SI) joint was not constant between stabilization techniques. The P/PMMA technique stabilized the SI joint such that no motion was appreciated at that articulation. The SI joint was not stabilized in the SOP and FACET specimens. It is unknown if preserving SI motion may have a clinical advantage. In the recommendations for SOP placement described above, bridging the SI joint is advised, so SI motion would be lost if this was performed. Anchoring implants to the ilium is suggested because of the historically poor screw purchase achieved in the sacrum alone.

Cantilever bending does typically result in a higher bending moment at the point of fixation of the specimen to the rigid stand compared to at the tip where the load is applied. In contrast, an even bending moment is applied with a 4-point fixture. The cantilever model was selected because it appeared to replicate the loads that would be applied to the LS region when the hind limbs were in stance phase and the load of the front half of the body was acting on the lumbar spine. Because the ilial shafts were potted right up to the sacrum, the highest bending moments would have been present on the SI joints and the LS joint. Of course, all models of spine motion are gross simplifications since they disregard the very important contribution of the active stabilizers of the spine.

The primary limitation of this study was that, because SI motion was much greater than expected, we were not able to compare the failure properties of the three different stabilization techniques. Another limitation of the study design was that the ability of the fixation methods to resist lateral bending and axial rotational forces was not evaluated.

Conclusion

This cadaver study demonstrated that stabilization of the lumbosacral junction by all three of the applied fixation techniques leads to decreased motion in flexion and extension. It is unknown whether the stabilization techniques used in combination with the dorsal laminectomy and partial discectomy procedures will provide sufficient stability for healing in clinical cases.

Supplemental Information

Supplemental Information 1 LS stabilization data se

LS stabilization data set

Click here for additional data file.

We thank Kristal Wilson for her assistance in acquiring data in the bioengineering laboratory and Robert Thatcher for his assistance with the figures.

Additional Information and Declarations

Competing Interests

Author Contributions

Animal Ethics

Peter Early and Simon Roe were once paid an honorarium as a course instructor by Orthomed, producer of the SOP Locking Plate system.

Peter Early conceived and designed the experiments, performed the experiments, analyzed the data, contributed reagents/materials/analysis tools, wrote the paper, prepared figures and/or tables, reviewed drafts of the paper.

Peter Mente conceived and designed the experiments, performed the experiments, analyzed the data, contributed reagents/materials/analysis tools, reviewed drafts of the paper.

Stacy Dillard performed the experiments, contributed reagents/materials/analysis tools, reviewed drafts of the paper.

Simon Roe conceived and designed the experiments, analyzed the data, wrote the paper, prepared figures and/or tables, reviewed drafts of the paper.

The following information was supplied relating to ethical approvals (i.e., approving body and any reference numbers):

This was a cadaver study and the dogs used for this study had been previous humanely euthanasied for reasons not related to the study. No IACUC was needed.

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
