# Peer review of "In vitro biomechanical evaluation of internal fixation techniques on the canine lumbosacral junction"

_PeerJ, doi:10.7717/peerj.1094_

## Round 0.1 · original submission · Major Revisions

Please, carefully considered all the suggestions made by the reviewers in the revised version of your manuscript.

·

Basic reporting

Figure 1 needs to have the identifying marks changed (the size difference and color difference of the circles is not distinct enough) - the authors should provide a new figure with different identifiers.

Experimental design

The method is inadequately described and the paper would benefit from more explanation of the method and a diagram or labelled photograph of the experimental jig.
The authors should include more detail of the method to allow the reader to recreate the experimental conditions.
A statistical test has been performed but the statistical methodology has not been described in the M&M

Validity of the findings

The data is novel and of interest to surgeons however the conclusions that are made are limited in their impact.
The differences in mode of failure and mechanical load at failure between the constructs could be better evaluated and discussed.
The final conclusion that all constructs render the lumbosacral junction significantly stiffer than a native spine after dorsal laminectomy and discectomy is correct but there appears to be an advantage to the P/PMMA construct.

Additional comments

Reviewer comments:

Abstract

The abstract is not truly representative of the paper as it does not indicate the differences at failure. In reading the paper I would conclude that P/PMMA is mechanically superior.
The following sentence needs rewording:
The range of motion (ROM) with 3 Nm of flexion and extension was determined by monitoring movement of L7 in a cantilever bending model.

Line 72 The authors use the terms subluxation of the sacrum and lumbosacral instability

Whilst these terms are commonly used by other authors, there is very little actual quantification of instability. The term instability implies increased motion at the lumbosacral junction, but there most studies indicate that as DLSS progresses the range of motion in flexion-extension actually decreases. Subluxation implies loss of articular process joint apposition. Some dogs do display displacement of the sacrum ventral to the correct articulation with L7, but these are in the minority. Perhaps the authors need to justify the concept of dorsal stabilisation as it applies to disc degeneration and DLSS.

75 -76 Surgical management is recommended for patients with severe or recurrent pain that is not responsive to medical management or when neurologic deficits are present. This needs referencing.

Surgical decision making in DLSS is currently based on expert opinion rather than hard scientific fact. The authors should indicate that controversy exists as to whether to decompress (dorsal laminectomy and/or foraminotomy) or stabilise (dorsal implants). The authors should discuss the recent paper by Jeffery et al 2014. Those authors call for studies of DLSS to be multi-centred with standardised inclusion criteria, randomisation to treatment groups and blinded evaluators using outcome measures (Jeffery et al. 2014). Jeffery ND, Barker A, Harcourt-Brown T. What progress has been made in the understanding and treatment of degenerative lumbosacral stenosis in dogs during the past 30 years? The Veterinary Journal 201, 9–14, 2014

89 Material and Method – should this be Materials and Methods?
• I think a photograph of the jig set-up, or at least a schematic drawing of how the experiment was conducted would be very useful.
• I also feel that the methodology is not sufficiently detailed to allow a reader to be able to repeat the testing conditions.
• The simple cantilever bending model should be better justified in the discussion.
• Was there a reason to use 5-hole plates for the SOP constructs given that only the first 3 pearls appear to have been used (from the figure).
• If some dogs actually had screws in holes 1 and 4, or 1 and 5 then this needs to be stated/described. The length of gap between the screws will alter the biomechanics.

91 ….skeletally mature non chondrodystrophic dogs with weights ranging from 23.6-36.7 kg.
The weight data should include a median as well as a range.

97 Following the ROM analysis, one of the three dorsal fixation technique was applied to the specimen

101 ROM of native specimen’s was 32.8±6.4° and after laminectomy and discectomy to 40.2±5.6°. This sentence does not make sense to me

102 Positive profile pins (4.0mm diameter) were used – This introduces a significant difference between the constructs. In addition to the extra 2 screws (6 not 4 for SOP) the shaft diameter is greater therefore the pins are much stronger than the 3.5mm cortical screws. This difference and the advantage of the PMMA construct as created, should be noted it the discussion.

103 Two pins were placed into the pedicle of L7, two pins into the sacrum and two pins into the ilium.
If you used the landmarks established by Meij et al, then you should cite Meij BP, Suwankong N, Van Der Veen AJ, Hazewinkel HAW. Biomechanical flexion-extension forces in normal canine lumbosacral cadaver specimens before and after dorsal laminectomy-discectomy and pedicle screw-rod fixation. Veterinary Surgery 36, 742–51, 2007

105-6 PMMA was applied, incorporating all of the pins. The PMMA was allowed to harden for a minimum of 1 hour before testing.
You should describe if the PMMA was in contact with the bone surface of L7 and the sacrum – i.e. conformed to the shape of the

112 Implant entry points for the three fixation techniques are identified in Fig 1

I do not find the identifying circles very easy to interpret. The size difference and different grey used is not enough to distinguish the alternative implant positions clearly. Could crosses be used instead of circles on one side?

124-7 After each fixation technique was applied the ROM of the stabilized specimens was significantly decreased (p<0.001) compared to ROM after dorsal laminectomy and discectomy (mean of all specimens for all three groups = 40.2±5.6° (Early et al., 2013).
I am not sure why there is a reference here, these are your own data, you refer to Early et al in the M&M so no need to refer to this again unless you are citing that studies data.
The authors have not described the statistical method in the M & M.

130 One of the SOP specimen failed by loosening of the screws in L7 with 12.7 Nm applied while in extension.
In my own experiments with SOP in a bending model to failure (elastic limit) the screws bent in the section of unprotected screw between the SOP plate and the bone surface. Did this not occur in your model?

Discussion
141-2 These studies evaluated the lumbosacral spine segments using 4-point bending. In this study, a cantilever system was used as it was easier to
instrument and load the spinal segment.
I am no engineer so not qualified to comment, but can you take advice on how important a 4-pt bending mode would be?

142-5 The recently developed SOPTM Locking Plate System combines the advantages of a fixed angle stabilization system, like the P/PMMA, with lower bulk. The SOPTM plate can be contoured so that the locked screws are directed into the limited available bone stock.
It should be pointed out that the Manufacturers of the SOP (Orthomed UK) recommend 2 screws in each plate section per vertebrae either side of the area to be stabilised. Thus in the recommended model there would be 2 screws per pedicle either side of L7 and 2 screws in the left and right sides of the sacrum.
However, dimensionally this is not possible in all but the largest of patients. The authors should explain why they had to diverge from recommended practice (size and orientation of the pedicles of L7.

149 The SI joint was not stabilized in the SOP and FACET specimens and motion was observed but not quantitated at the SI joint in these specimens. Because SI motion resulted in movement of the L7-S1 complex, and the attached potentiometer, the specimen reached the maximum allowed motion (10 °) at bending loads that would not be expected to fail the implants.
The use of cameras might have allowed more observational data. The FACET specimens would be expected to pivot about the articular process joints in flexion and extension. The SOP constructs are well above the bone surface and as such there is a moment arm about which each screw can bend. The P/PMMA constructs are

152-5 We were not able to compare the failure properties of the three different stabilization techniques, because of the study design in regard to the motion at the SI joint. Another limitation of the study design was that the ability of the fixation methods to resist lateral bending and axial rotational forces was not evaluated.

Anecdotally I have used SOP in exactly the conformation you describe and had screw failures within 3 months of implantation in German shepherd working dogs.
The data you describe would indicate to me that both the FACET and SOP constructs failed at much lower bending stress that the P/PMMA. You could make more of this observation and try to explain it. For instance the PMMA, if conformed to the shape of the vertebra acts as a buttress against the bone in addition to having thicker and more fixation implants.

Reviewer 2 ·

Basic reporting

This manuscript is well written, with just a few spelling and grammatical errors.

On line 125, there should be a comma between "applied" and "the".
On lines 128 and 129, it should read "specimens" not "specimen".
Line 151-reword; the phrase "failed the specimens" is grammatically incorrect and sounds awkward. Suggest using something like "caused implant failure."

Experimental design

The experimental design is straightforward. I think it is important to compare the biomechanical data from the constructs to the intact specimen data, not just those data from laminectomy/discectomy specimens. It seems that data from intact specimens were gathered, just not mentioned. Also, I think that it would be valid to compare ROM among the constructs during repetitive cycling, as long as it is explained that differences may be-in part-due to differences in motion at the SI joint. Similarly-for failure-it would be worthwhile to look at the comparative stiffness of the P/PMMA construct vs. the other two constructs. This may make the P/PMMA construct more attractive in clinical situations in which SI instability is a factor (something for the discussion section).

Validity of the findings

The findings are valid. However, it is unclear as to what extent the dorsal laminectomy/discectomy procedure increased ROM compared to the intact specimens (the premise for considering stabilization for DLS) and whether and to what degree any of the fixation constructs normalized or improved the intact condition of the specimens. The introduction and discussion sections are very cursory. For the introduction section, I suggest the authors note that fracture/luxation is another reason to consider such fixation constructs. In the discussion, it would also be worth mentioning that one of the constructs (P/PMMA) may also have utility in managing concurrent SI fracture/luxation.

Additional comments

This study was well done, but there are a few things that would make the paper stronger.

---

## Round 0.2 · accepted · Accept

Thank you for publishing with us.

·

Basic reporting

Re-review

The points I made in my previous review have been addressed. I am now happy for the paper to be accepted

Experimental design

Better described and reasonable design

Validity of the findings

conclusions are now valid and measured

Additional comments

Thank you for making the changes which have improved the manuscript